# A Bearing Fault Diagnosis Method Based on Wavelet Denoising and Machine Learning

**Shaokun Fu** [1], **Yize Wu** [1], **Rundong Wang** [2] **and Mingzhi Mao** [3,*]

1   Information and Computing Science, China University of Geosciences, Wuhan 430074, China;
    20201004172@cug.edu.cn (S.F.); wuyize@cug.edu.cn (Y.W.)
2   Computer Science and Technology, China University of Geosciences, Wuhan 430074, China;
    wangrd@cug.edu.cn
3   Mathematics and Physics, China University of Geosciences, Wuhan 430074, China
*   Correspondence: mingzhi-mao@163.com

**Abstract:** There are a lot of interference factors in the operating environment of machinery, which makes it ineffective to use traditional detection methods to judge the fault location and type of fault of the machinery, and even misjudgment of the fault location and type may occur. In order to solve these problems, this paper proposes a bearing fault diagnosis method based on wavelet denoising and machine learning. We use sensors to detect the operating conditions of rolling bearings under different working conditions to obtain datasets of different types of bearing failures. On the basis of using the wavelet denoising algorithm to reduce noise, we comprehensively evaluated five machine learning models, including K-means clustering, decision tree, random forest, and support vector machine to classify bearing faults and compare their results. By designing the fault classification evaluation prediction criteria, the following conclusions are drawn. The model proposed in this paper is significantly better than other traditional diagnostic models for bearing faults. In order to solve the problem of weak signal strength and background noise interference, this paper selects a better noise reduction algorithm under different quantitative evaluation indicators for wavelet denoising, which can better restore the true characteristics of the fault signal. Using unsupervised learning and supervised machine learning classification algorithms, the evaluation indicators before and after denoising are compared to make the classification results more accurate and reliable. This article will help researchers to intelligently diagnose the faults of rolling bearing equipment in rotating machinery.

**Keywords:** fault detection; wavelet denoising; K-means clustering; decision trees; random forests; support vector machines; AdaBoost algorithm

## 1. Introduction

With the rise of global industrialization and the widespread use of machinery, fault data have become increasingly dense and unrelated. The goal of fault diagnosis technology for instruments and equipment is to monitor the operational state of machines and determine their overall or partial functionality. By employing data mining and machine learning techniques, faults and their underlying causes can be identified, and future fault developments can be predicted.

The technical characteristics of fault type identification are as follows: reducing the impact of environmental noise or abnormal data on original data, extracting reliable waveform feature criteria, and selecting or improving existing machine learning methods. The work of [1–3] has received long-term attention in the industry [4]. As a key component of machinery, bearings can fix objects and provide loads [5]. If the bearing fails, it will lead to a decline in the quality of the product and cause serious economic losses [6]. So bearing fault detection is the focus and difficulty of current research [7]. Abdul et al. summarize

recently published work on the application of artificial intelligence to bearing fault detection in rotating machinery [8,9]. Scholars have proposed many intelligent fault detection methods for the pulse vibration signals generated by bearings. Among these methods, the bearing detection method of deep learning is used the most. Compared with the traditional bearing detection method, deep learning can accurately identify fault information in the context of big data, and reduces the requirement for professional knowledge [10]. Zhang et al. improved the feature extraction method of the convolutional neural network and performed fault diagnosis on bearings.

Although the above methods have achieved great success, there is a strong dependence on the fault data, and more labels are required when training the model. Such research is contrary to the real industrial environment: in the real industrial environment, there are far more normal data than fault data. Such a dataset will greatly reduce the training effect of the deep learning model. Ref. [11] has attracted widespread attention in the academic community.

In order to solve the above-mentioned problems in the industry, Ref. [12] Sun et al. used traditional machine learning, an artificial neural network, deep learning and transfer learning to carry out a typical fault diagnosis method of a wind power system based on the ML method from the theoretical basis and industrial application. An overview is given and its advantages and disadvantages are discussed [13]. He et al. worked on improving the model structure and classification loss function, proposing to optimize support vector machines with dynamic penalty factors [14]. Praneeth Chandra et al. used unsupervised machine learning and the DBSCAN algorithm to detect defects on rail fixtures [15]. Praneeth Chandra et al. diagnosed defects using supervised machine learning. The learning data are acquired by sensors, and six machine learning models are used. The result values are best when using the the AdaBoost model [16]. Yoon et al. found unexpected test accuracy in cross-validation for machine learning [17]. Yang et al. proposed a new machine learning model for automatic labeling, feature extraction, model building, model fusion, and evaluation of railway data [18]. Nason et al. described a stationary wavelet transform and used it in astronomy as well as veterinary anatomy [19]. Baydar et al. used wavelet transform to compare acoustic signals and vibration signals, and the results showed that acoustic signals have a strong influence on early fault detection [20]. Firas et al. used an optimized stationary wavelet packet transform to provide identification for detecting IM bearing faults at an early stage and verified the effectiveness of the method experimentally [21]. Lu et al. proposed a higher-density wavelet packet neighborhood coefficient denoising algorithm, and experiments showed that the algorithm can significantly improve the signal-to-noise ratio [22]. Aslam et al. proposed a fault detection method based on SNR [23]. Shim et al. used the improved CNN LeNet-5 model to detect railway anomalies. Ref. [24] M. F. Palangar et al. trained a classifier based on two indicators of experimental data and confirmed the reliable performance of ADD [25]. Stępień K et al. present a method for evaluating the surface texture of machine parts using wavelet transform of three selected indices.

This paper proposes a fault type classification and identification model based on wavelet denoising and machine learning. There are a lot of interference factors in the operating environment of machinery, which makes it ineffective to use traditional detection methods to judge the fault location and type of fault of the machinery, and even misjudgment of the fault location and type may occur. Therefore, how to realize intelligent fault diagnosis in an environment full of a large number of interference factors is an urgent problem to be solved. We use two methods of wavelet soft threshold denoising and wavelet packet analysis to denoise the original fault signal data. Eliminate the interference factors in the environment, check the correctness of the denoising results, and select the denoising method with better effect on the basis of quantitative indicators. Based on the wavelet denoising algorithm, five of the most widely used machine learning models are applied: K-means clustering, quadratic support vector machine (SVM), random forest (RF), decision tree (DT) and AdaBoost algorithm. These machine learning algorithms are used for binary

classification of bearing faults and to compare their results. By designing the fault classification evaluation prediction criteria, the following conclusions are drawn. If unsupervised machine learning methods are used, K-means clustering can achieve good classification results; if supervised machine learning methods are used, decision tree classification can achieve the best results. In addition, the focus of our future work is as follows: First, increase the types of rolling bearing faults, narrow the characteristic differences of fault signals, and improve the adaptability and flexibility of the model. Second, more types of faults and larger datasets are considered to improve the generalization ability of the model. Finally, the wavelet denoising algorithm can be further optimized by selecting different wavelet bases, decomposition levels and threshold functions according to different types of signals. Additionally, machine learning models can further improve their performance by tuning parameters, selecting features, or combining multiple models. In this paper, it is of great significance to intelligently diagnose the faults of instruments and equipment.

## 2. Methods

### 2.1. Wavelet Soft Thresholding Denoising

Wavelet transform can process the signal in a more subtle way, and can also better express some characteristics of the signal. There are also many denoising methods in wavelet analysis, and these different denoising methods will also have a great impact on the denoising effect of each signal. The Wavelet threshold denoising method is the most widely used method in engineering. The basic idea is as follows: a noise-containing one-dimensional signal model can be expressed as $f(t) = s(t) + n(t)$, where $s(t)$ is the original signal, and $n(t)$ is Gaussian white noise with variance $\sigma^2$, obeying $N(0, \sigma^2)$. Wavelet transform belongs to linear transformation, and the wavelet coefficient obtained after discrete wavelet transform of $f(t)$ is divided into two parts: signal and noise. The wavelet coefficients with larger amplitudes are useful signals. The wavelet coefficient with smaller amplitude is the energy of the noise signal. Therefore, an appropriate threshold $\lambda$ can be found for distinguishing noise and signal, and the useful signal is retained with reference to this threshold. For this problem, we use the soft threshold method [26] for denoising, first transforming the actual signal into the wavelet domain using wavelet transform. Then the wavelet coefficients are processed by the nonlinear shrinkage rule; finally, the thresholded wavelet coefficients are subjected to inverse wavelet transform to obtain the denoising signal. The soft threshold function is defined as follows:

$$\bar{w}_{i,j} = \begin{cases} \text{sign}\left(w_{j,k}\right)\left(\left|w_{j,k}\right| - \lambda\right) & \left|w_{j,k}\right| \geq \lambda \\ 0 & \left|w_{j,k}\right| < \lambda \end{cases} \tag{1}$$

### 2.2. Wavelet Packet Analysis Denoising

The idea of wavelet packet analysis and wavelet soft threshold denoising [27] is basically the same, but the analysis method of wavelet packet is more complex and flexible. When analyzing, it decomposes the low-frequency part and high-frequency part of the previous level at the same time, making the local analysis capability more accurate. The wavelet packet decomposition also needs to choose the wavelet packet base and level of decomposition. For the selection of the wavelet packet base, it is necessary to choose the wavelet packet base with better symmetry and regularity. The wavelet with better symmetry does not produce phase distortion, and it is easy to obtain a smooth reconstructed signal with the wavelet with better regularity. In the wavelet base family, the Symlets family is similar to the Daubechies family, which also has orthogonality, biorthogonality, compact support and approximate symmetry, and can perform discrete wavelet transform. Therefore, in the part of wavelet packet analysis and denoising, the wavelet packet base of this family is used for denoising.

### 2.3. Wavelet Denoising Index

We used wavelet soft threshold denoising and wavelet packet analysis denoising to perform denoising respectively and selected six indicators: mean square error (*MSE*), sum square error (*SSE*), root mean square error (*RMSE*), coefficient of determination ($R^2$), mean absolute error (*MAE*) and signal-to-noise ratio (*SNR*) to evaluate the denoising effect. Among these six evaluation indexes, except for the coefficient of determination ($R^2$) and *SNR* which are maximum indexes, the other four indexes are minimum indexes. In addition to quantitative evaluation indicators, the denoising effect can also be intuitively displayed by comparing the original signal with the denoised signal image.

Let $y_i$ be the sampling point of the original signal, $\hat{y}_i$ be the sampling point of the signal after denoising, and $m$ be the number of sampling points.

$$MSE = \frac{1}{m}\sum_{i=1}^{m}(y_i - \hat{y}_i)^2 \tag{2}$$

$$SSE = \sum_{i=1}^{m}(y_i - \hat{y}_i)^2 \tag{3}$$

$$RMSE = \sqrt{\frac{1}{m}\sum_{i=1}^{m}(y_i - \hat{y}_i)^2} \tag{4}$$

$$R^2 = 1 - \frac{\sum_{i=1}^{m}(y_i - \hat{y}_i)^2}{\sum_{i=1}^{m}(y_i - \bar{y}_i)^2} \tag{5}$$

$$MAE = \frac{1}{m}\sum_{i=1}^{m}|y_i - \hat{y}_i| \tag{6}$$

$$SNR = 10 \times \lg\left[\frac{\sum_{i=1}^{m}y_i^2}{\sum_{i=1}^{m}(y_i - \hat{y}_i)^2}\right] \tag{7}$$

### 2.4. K-Means Clustering Algorithm

The strategy of K-means clustering is to select the optimal partition by minimizing the loss function. The optimal partition is selected by minimizing the loss function.

First use the square of the Euclidean distance as the distance between samples:

$$d(x_i, y_i) = \sum_{k=1}^{m}\left(x_{ki} - x_{kj}\right)^2 = \left\|x_i - x_j\right\|^2 \tag{8}$$

Then the sum of the distances between the sample and the center of the class to which it belongs is the loss function, namely,

$$W(C) = \sum_{i=1}^{k}\sum_{c(i)=1}\left\|x_i - \bar{x}_l\right\|^2 \tag{9}$$

where $\bar{x}_l = \left(\bar{x}_{1l}, \bar{x}_{2l\dots}\bar{x}_{nl}\right)^T$ is the mean center of classes, namely,

$$n_l = \sum_{i=1}^{n}I(C(i) = l) \tag{10}$$

It is an indicator function, which takes a value of 1 or 0. The function $W(C)$, also known as energy, represents the similarity of samples in the same class.

Therefore, K-means clustering can be transformed into solving an optimization problem:

$$C^* = \arg\min_{C}W(C) \tag{11}$$

The flow chart of the steps of the K-means clustering algorithm is shown in Figure 1.

**Figure 1.** 5-layer decomposition of original signal and denoised signal.

### 2.5. Support Vector Machine (SVM)

Support vector machines map vectors into a higher dimensional space and, for linearly separable tasks, find a hyperplane with the largest margin. Two parallel hyperplanes are built on both sides of the hyperplane separating the data, and the separating hyperplane maximizes the distance between the two parallel hyperplanes. The larger the distance or gap between parallel hyperplanes, the smaller the overall error of the classifier. The steps are as follows:

- Import data;
- Normalize the data;
- Execute the support vector machine to find the optimal hyperplane;
- Draw classification hyperplane kernel support vector;
- Using polynomial features to perform support vector machines in high-dimensional space;
- Select an appropriate kernel function and execute a nonlinear support vector machine.

### 2.6. Decision Tree (DT)

Decision tree is a typical supervised classification method that approximates discrete function values to classify data. First, it processes data and uses an inductive algorithm to generate readable rules and decision trees. Then, it uses the decision to analyze new data. Essentially, decision trees classify data through a series of rules.

### 2.7. Random Forest (RF)

As an idea of integrated learning, random forest inputs the data obtained by random sampling into many decision trees, performs voting, and obtains the final output result.

Constructing a random forest requires the following four steps:

- A sample with a sample size of N is drawn N times with replacement, and one sample is drawn each time, then N samples can be formed. These N samples are used to train a decision tree as samples of the root node of the decision tree;
- If each sample has M attributes, when each node of the decision tree needs to be split, randomly select m attributes from the M attributes to meet the condition $m \ll M$. From these attributes, information gain is used to select an attribute as the splitting attribute of the node;
- During the formation of the decision tree, each node is split according to step 2;
- Build a large number of decision trees according to steps 1–3, and then construct the entire random forest.

### 2.8. AdaBoost Algorithm

The AdaBoosting algorithm [28] is an important ensemble learning technique, which can enhance a weak learner whose prediction accuracy is only slightly higher than random guessing into a strong learner with high prediction accuracy, which provides a new idea for solving the difficulty of directly constructing a strong learner. AdaBoost is one of the most successful algorithms.

A brief description of the AdaBoost algorithm is as follows:

- First initialize the weight distribution D1 of the training data. When there are *N* training sample data, each training sample is given the same weight at the beginning: $w1 = \frac{1}{N}$;
- Then, train the weak classifier H. In the specific training process, if a certain training sample point is accurately classified by the weak classifier H, then its corresponding weight should be reduced when constructing the next training set. Conversely, if a training sample point is misclassified, its weight should be increased. The sample set whose weights have been updated is used to train the next classifier, and the entire training process will continue to iterate.
- Finally, combine the weak classifiers obtained from each training into a strong classifier. After the training process of each weak classifier is over, the weight of the weak classifier with a small classification error rate is increased so that it plays a greater decisive role in the final classification function. The weight of the weak classifier with a large classification error rate is reduced so that it plays a less decisive role in the final classification function.

### 2.9. Clustering Evaluation Index

When evaluating the prediction results, select the following six indicators: accuracy rate, recall rate, time-consuming, contour coefficient, F value and entropy value. The calculation formulas of precision rate, recall rate and contour coefficient index are as follows:

There are four situations in the data test results: *TP*: predicted positive, actually positive; *TN*: predicted negative, actually negative; *FP*: predicted positive, actually negative; *FN*: predicted negative, actually positive.

The calculation formula of the indicator is as follows:

- Accuracy:

$$Accuracy = \frac{TP + TN}{TP + TN + FN + FP} \tag{12}$$

- Recall rate:

$$R = \frac{TP}{TP + FN} \tag{13}$$

- Silhouette coefficient:

$$S(i) = \begin{cases} 1 - \frac{a(i)}{b(i)} & a(i) < b(i) \\ 0 & a(i) = b(i) \\ \frac{b(i)}{a(i)} - 1 & a(i) > b(i) \end{cases} \tag{14}$$

where $a(i)$ represents the cohesion of the sample point, and the calculation method is as follows:

$$a(i) = \frac{1}{n-1} \sum_{\substack{j \neq i}}^{n} \text{distance}(i, j) \tag{15}$$

where $j$ represents other sample points in the same class as sample $i$, and distance represents the distance between $i$ and $j$. So the smaller $a(i)$ is, the closer the class is. $b(i)$ is calculated similarly to $a(i)$. However, it is necessary to traverse other clusters to obtain multiple values $\{b_1(i), b_2(i), b_3(i) \ldots b_m(i)\}$ and select the smallest value as the final result.

- The entropy value of the entire cluster division is:

$$e = \sum_{i=1}^{K} \frac{m_i}{m} e_i \tag{16}$$

where $K$ is the number of clusters, $m$ is the number of members involved in the overall cluster division, $m_i$ is the number of all members in cluster $i$, and $e_i$ is the entropy value of each cluster:

$$e_i = -\sum_{j=1}^{L} P_{ij} \log_2 P_{ij} \tag{17}$$

In this formula, $P_{ij}$ refers to the probability that a member in cluster $i$ belongs to class $j$,

$$P_{ij} = \frac{m_{ij}}{m_i} \tag{18}$$

$m_{ij}$ is the number of members in cluster $i$ belonging to class $j$.

- The $F$ value is the weighted proportion of different categories of data in the total number of samples. Its formula is as follows:

$$F = \frac{n_i}{n} F_i + \frac{n_j}{n} F_j \tag{19}$$

where $n_i, n_j$ is the number of samples of each category, and $F_i, F_j$ is the $F$ value of each type of sample:

$$F_i = \frac{2 \cdot P_i \cdot R_i}{P_i + R_i} \tag{20}$$

$P_i$ is the precision rate, and its calculation formula is:

$$P = \frac{TP}{TP + FP} \tag{21}$$

Use the above formula to calculate each evaluation index.

## 3. Results and Discussion

### 3.1. Dataset Introduction

We have measured the fault data of rolling bearing wheel sets in a factory in Wuhan. The operating status of rolling bearings in rotating machinery includes normal, inner ring faults and outer ring faults. The inner ring fault and the outer ring fault are recorded as type A and type B faults, respectively. Consider the FFT transformation, choose 2048 as the sample sampling length, and make sample sets in an overlapping manner. In the training

set, the normal data contain 200 samples, and each fault dataset of class A and class B contains 100 samples.

### 3.2. Display of Various Index Results of Different Noise Reduction Methods

#### 3.2.1. Index Analysis of Wavelet Soft Threshold Denoising

When performing wavelet soft threshold denoising, it is necessary to set multiple parameters, including threshold selection criteria, wavelet basis functions, and wavelet levels of decomposition. Among the wavelet basis functions of Haar, Daubechies, Coiflets and other wavelet basis functions of the wavelet basis family, the wavelet basis of the Daubechies family has orthogonality, biorthogonality, compact support, and approximate symmetry, and can perform discrete wavelet transform. Therefore, wavelet transformation is performed using this wavelet basis function. At the same time, wavelet bases with different vanishing moments in the Daubechies family are used to process the denoised signal, and the processing results are compared. In practice, the basic wavelet is often not only required to meet the admissible condition, but also to impose the so-called vanishing moment condition, so that as many wavelet coefficients as possible are zero or as few non-zero wavelet coefficients as possible are generated, which is conducive to data compression and noise elimination. The larger the vanishing moment, the more wavelet coefficients are zeroed out. But in general, the higher the vanishing moment is, the longer the support length is, and the smoother the wavelet is. Therefore, in terms of support length and vanishing moment, we must make a compromise. Considering the general situation, the selected level of decomposition is 4. Change the vanishing moments of the Daubechies wavelet family, and take db3, db6, and db9 three vanishing moments for noise reduction, and analyze the influence of different vanishing moments on noise reduction.

Tables 1 and 2 show the index analysis results of the first fault signal of type A and type B after wavelet soft threshold denoising.

**Table 1.** Analysis results of the first signal index of Type A fault.

| Type of Wavelet | MSE | SSE | RMSE | $R^2$ | MAE | SNR |
|---|---|---|---|---|---|---|
| db3 | $1.20 \times 10^3$ | $5.04 \times 10^0$ | $3.50 \times 10^{-2}$ | $9.88 \times 10^{-1}$ | $4.46 \times 10^{-6}$ | $1.96 \times 10^1$ |
| db6 | $6.40 \times 10^{-4}$ | $2.61 \times 10^0$ | $2.52 \times 10^{-2}$ | $9.94 \times 10^{-1}$ | $2.59 \times 10^{-6}$ | $2.25 \times 10^1$ |
| db9 | $4.60 \times 10^{-4}$ | $1.89 \times 10^0$ | $2.15 \times 10^{-2}$ | $9.96 \times 10^{-1}$ | $2.30 \times 10^{-5}$ | $2.39 \times 10^1$ |

**Table 2.** Analysis results of the first signal index of Type B fault.

| Type of Wavelet | MSE | SSE | RMSE | $R^2$ | MAE | SNR |
|---|---|---|---|---|---|---|
| db3 | $3.87 \times 10^{-3}$ | $1.59 \times 10^1$ | $6.22 \times 10^{-2}$ | $8.64 \times 10^{-1}$ | $3.56 \times 10^{-6}$ | $8.69 \times 10^0$ |
| db6 | $1.91 \times 10^{-3}$ | $7.82 \times 10^0$ | $4.37 \times 10^{-2}$ | $9.33 \times 10^{-1}$ | $4.72 \times 10^{-6}$ | $1.18 \times 10^1$ |
| db9 | $1.33 \times 10^{-3}$ | $5.47 \times 10^0$ | $3.65 \times 10^{-2}$ | $9.53 \times 10^{-1}$ | $5.06 \times 10^{-6}$ | $1.33 \times 10^1$ |

#### 3.2.2. Wavelet Packet Analysis Denoising Index Analysis

The level of decomposition has a great influence on the denoising effect. If there are too many levels of decomposition, all the wavelet space coefficients of each level will be processed. It is easy to cause serious information signal loss, decrease in signal-to-noise ratio after denoising, and slow processing due to increased computation. If the level of decomposition is too small, the denoising effect is not ideal. For each signal, there is a level of decomposition that denoises best or is close to best. Therefore, when dealing with this problem, on the premise that the wavelet packet base and its vanishing moment are fixed, different levels of decomposition are selected, and the denoising effects are compared to find the most suitable level of decomposition for the denoising signal. Select the Symlets wavelet family, fix its vanishing moment to 6, change its level of decomposition, and perform four-level of decomposition and five-level of decomposition, respectively.

Tables 3 and 4 show the index analysis results of the first fault signal of type A and type B after denoising by wavelet packet analysis.

**Table 3.** Analysis results of the first signal index of Type A fault.

| Levels | MSE | SSE | RMSE | $R^2$ | MAE | SNR |
|---|---|---|---|---|---|---|
| 4 | $6.25 \times 10^{-4}$ | $2.56 \times 10^0$ | $2.50 \times 10^{-2}$ | $9.94 \times 10^{-1}$ | $4.46 \times 10^{-6}$ | $2.26 \times 10^1$ |
| 5 | $1.10 \times 10^{-2}$ | $4.50 \times 10^1$ | $1.05 \times 10^{-1}$ | $8.95 \times 10^{-1}$ | $4.46 \times 10^{-6}$ | $1.01 \times 10^1$ |

**Table 4.** Analysis results of the first signal index of Type B fault.

| levels | MSE | SSE | RMSE | $R^2$ | MAE | SNR |
|---|---|---|---|---|---|---|
| 4 | $1.88 \times 10^{-3}$ | $7.74 \times 10^0$ | $4.35 \times 10^{-2}$ | $9.34 \times 10^{-1}$ | $5.06 \times 10^{-6}$ | $1.18 \times 10^1$ |
| 5 | $2.67 \times 10^{-2}$ | $1.09 \times 10^2$ | $1.63 \times 10^{-1}$ | $6.50 \times 10^{-2}$ | $1.32 \times 10^{-6}$ | $3.00 \times 10^{-1}$ |

*3.3. Comparative Analysis of Noise Reduction Results*

For type A faults, the comparative analysis of wavelet threshold denoising shows that under the condition of constant level of decomposition, with the increase of vanishing moment, the indicators after denoising basically become better. However, when the vanishing moment increases to 9, the mean absolute error MAE increases sharply. It is considered that the signal after denoising changes greatly compared with the original signal, and some characteristics of the original signal may be lost, so it is not used. For wavelet packet analysis denoising, when the wavelet base is determined to be sym6, the denoising effect of wavelet packet analysis with four-level of decomposition is better than that with five-level of decomposition. Then we compare the denoising effects of wavelet soft thresholding and wavelet packet analysis. It is judged that the noise reduction effect of four-level of decomposition with wavelet base sym6 in wavelet packet analysis is better than that of four-level of decomposition with wavelet base db6 in wavelet soft threshold analysis.

Taking the four-level of decomposition whose wavelet base is sym6 in the first four signals of type A fault as an example, the denoising results of wavelet packet analysis are shown in Figure 2.

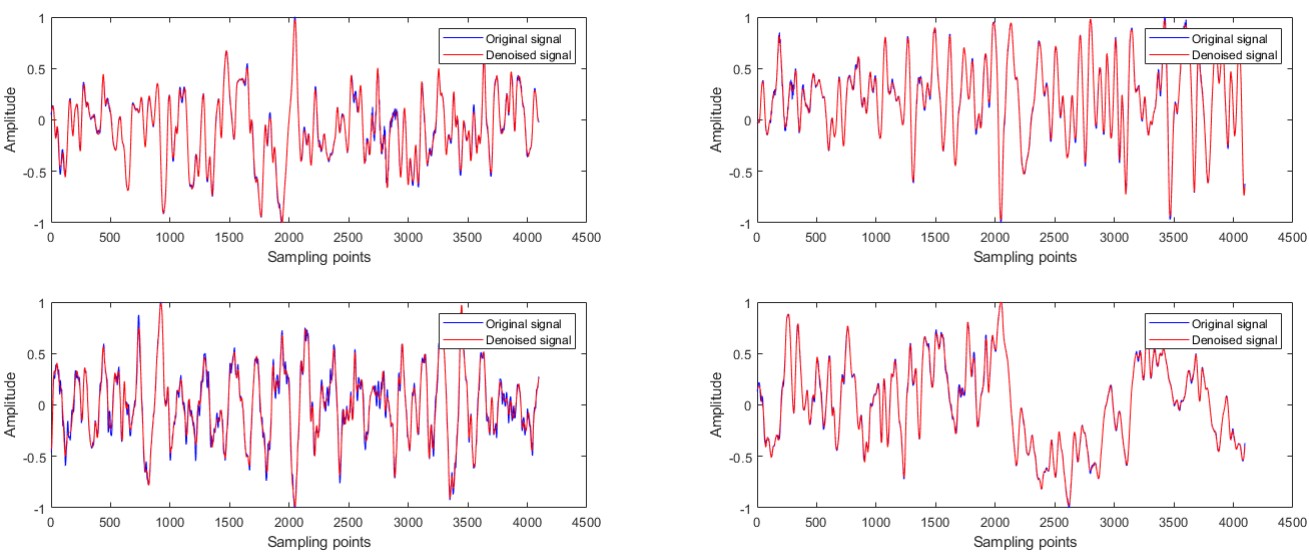

**Figure 2.** The original signal and denoised signal of the first four signals of Type A fault.

For the four signals of type B fault, the result analysis is basically consistent with the four signals of type A fault. However, in the comparison between wavelet soft threshold

and wavelet packet analysis, it is found that the noise reduction effect of 4-level of decomposition with wavelet base db9 in wavelet packet analysis is better than that of four-level of decomposition with wavelet base sym6 in wavelet soft threshold analysis.

Taking the four-level of decomposition with wavelet base as db9 in the four signals of Type B fault as an example, the denoising results of wavelet packet analysis are shown in Figure 3.

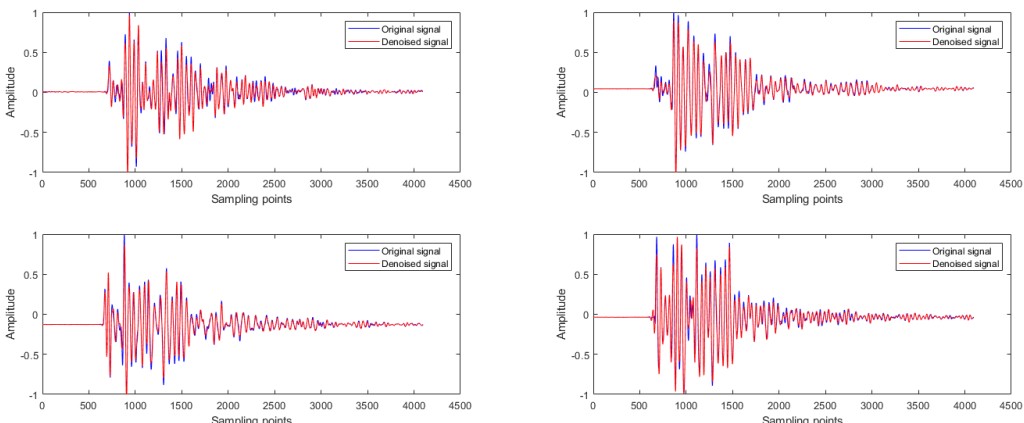

**Figure 3.** The original signal and denoised signal of the four signals of Type B fault.

It should be noted that, from the wavelet packet analysis results of the first signal of the Type B fault, it can be seen that the excessive level of decomposition has a serious impact on the denoising effect. From the SNR point of view, using the five-level of decomposition of the sym6 wavelet base, the signal-to-noise ratio is significantly lower than that of the four-level of decomposition, indicating that this denoising method loses a lot of useful signals. This can also be illustrated from the signal images before and after denoising, as shown in Figure 4.

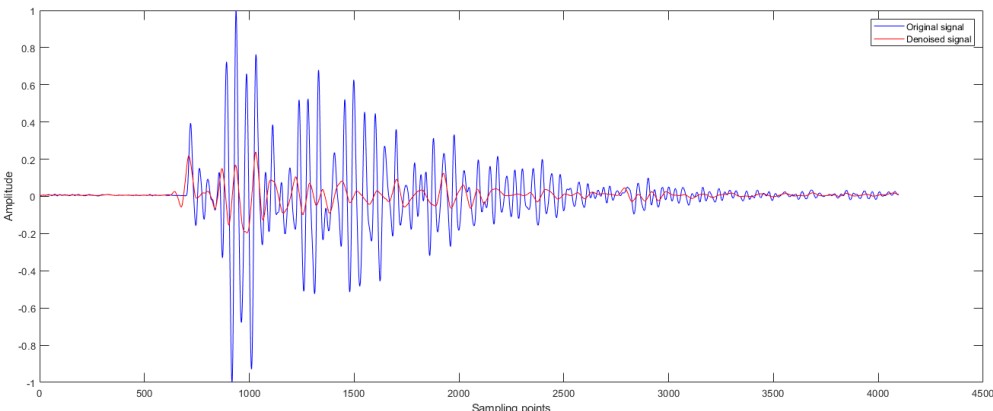

**Figure 4.** 5-level of decomposition of original signal and denoised signal.

### 3.4. Applications of Unsupervised Learning

For unsupervised clustering requirements, we use different scales of Shannon entropy $E5$, $E6$, $E7$, $E8$ and seven indicators of kurtosis, margin, and center of gravity frequency as clustering indicators for clustering. Among them, indicators $E5$, $E6$, $E7$, and $E8$ are energy indicators.

During data processing, it was observed that direct K-means clustering analysis on the original data is not effective, because K-means clustering only considers the absolute distance between each sample point and is not sensitive to data of different orders of magnitude. Therefore, it is considered to perform logarithmic transformation on the data

to expand the influence of the order of magnitude on the distance, so that it meets the requirements of cluster analysis. It is found that the clustering effect is better and the clustering result is stable. At this time, the effect diagram of K-means cluster analysis is shown in Figure 5.

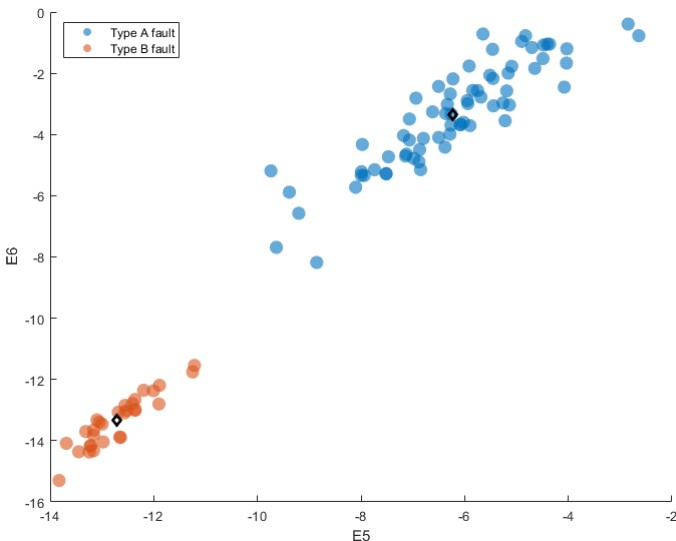

**Figure 5.** K-means cluster analysis effect diagram.

Table 5 shows the evaluation results of 100 K-means clustering.

**Table 5.** Evaluation index table of clustering results.

| Number of Experiments | Accuracy/% | Recall Rate/% | Time Cost/ms | Silhouette Coefficient | F Value | Entropy Value |
|---|---|---|---|---|---|---|
| 1 | 96 | 97.1 | 1.9 | 0.93 | 0.96 | 0.24 |
| 2 | 94 | 95.7 | 1.8 | 0.91 | 0.94 | 0.26 |
| ⋮ | ⋮ | ⋮ | ⋮ | ⋮ | ⋮ | ⋮ |
| 100 | 92 | 91.6 | 1.9 | 0.92 | 0.95 | 0.27 |
| average | 94 | 94.8 | 1.87 | 0.92 | 0.95 | 0.26 |
| standard deviation | 1.63 | 2.33 | 0.05 | 0.01 | 0.01 | 0.01 |

For this result, the mean accuracy was 94%, and each item achieved 90%. The clustering result is stable, and the standard deviation of accuracy is 1.63.

### 3.5. Applications of Supervised Learning

In order to solve the binary classification problem based on the supervised learning method, quadratic support vector machine (SVM), random forest (RF), decision tree (DT) and AdaBoost algorithm are used for binary classification.

On the basis of the evaluation index of unsupervised learning, the evaluation index of entropy value and contour coefficient is removed, and the evaluation index AUC value and precision rate are added. The AUC value is an evaluation index dedicated to the binary classification, which is used to evaluate the quality of the binary classification classifier. Its advantage is that it can avoid converting probability predictions into categories. A classifier with a larger AUC value has a higher correct rate.

Figure 6 shows the clustering results of the double SVM.

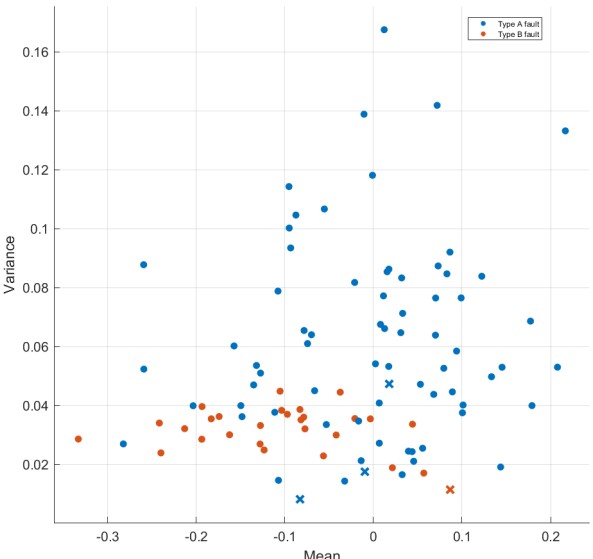

**Figure 6.** Double SVM clustering result graph.

Table 6 shows the results of classification evaluation metrics using various supervised machine learning algorithms.

**Table 6.** Evaluation index table of clustering results of different classifiers.

| Classification Method | Accuracy/% | Recall Rate/% | Time Cost/ms | Silhouette Coefficient | F Value | Entropy Value |
|---|---|---|---|---|---|---|
| decision tree | 96.7 | 96.7 | 18 | 0.968 | 0.966 | 0.987 |
| random forest | 96.7 | 96.7 | 1040 | 0.968 | 0.966 | 0.91 |
| Adaboost | 96.7 | 96.7 | 867 | 0.968 | 0.967 | 0.986 |
| double SVM | 96 | 95.8 | 1690 | 0.96 | 0.96 | 0.987 |

It can be seen from the table that the accuracy of double SVM classification is the lowest, and when the accuracy of other classification methods is the same, the time-consumption of the decision tree is the shortest and the AUC value is the highest. From this analysis, it can be concluded that for the binary classification algorithm using supervised learning, the effect of equipment fault diagnosis using decision tree classification is the most ideal.

Using the decision tree algorithm to conduct 100 classification experiments, the following evaluation Table 7 can be obtained:

**Table 7.** Evaluation table for 100 classification experiments.

| Number of Experiments | Accuracy/% | Recall Rate/% | Time Cost/ms | Silhouette Coefficient | F Value | Entropy Value |
|---|---|---|---|---|---|---|
| 1 | 96.7 | 96.7 | 18 | 0.968 | 0.966 | 0.987 |
| 2 | 97.2 | 97.2 | 18 | 0.973 | 0.974 | 0.987 |
| ⋮ | ⋮ | ⋮ | ⋮ | ⋮ | ⋮ | ⋮ |
| 100 | 96.4 | 96.4 | 18 | 0.965 | 0.962 | 0.987 |
| average | 96.8 | 96.8 | 18 | 0.969 | 0.967 | 0.987 |
| standard deviation | 0.33 | 0.33 | 0 | 0.003 | 0.005 | 0 |

For this result, the mean accuracy rate is 96.8%, and each item is above 95%. Therefore, the clustering result is stable, and the standard deviation of the accuracy rate is 0.330.

### *3.6. The Effect of Denoising on the Effect of Clustering Experiments*

In order to compare and analyze the impact of denoising on the test results, we counted the changes in the classification indicators of unsupervised machine learning and supervised machine learning, respectively.

It can be seen from Table 8 that if unsupervised learning is used for binary classification experiments, the accuracy and recall rate of the data after denoising will be greatly improved and the time spent will be shortened compared with that before denoising. It proves that the data features after denoising are more obvious than those before denoising. After noise reduction, the redundant data that need to be processed by the algorithm are reduced, so the usage time is reduced. The silhouette coefficient before noise reduction has a slight increase compared with that after noise reduction, which proves that the clustering effect is better after noise removal. The F value in the evaluation index indicates that the results of the test sample can represent the true degree of the whole. Because the data before and after denoising are clustered with all the signal samples, no test samples are proposed for learning, so the F value before and after denoising has not changed. The entropy value reflects the reliability of the information. A small decrease in the entropy value after denoising indicates that the original signal tends to be stable after removing the messy noise signal.

**Table 8.** Unsupervised clustering features of signals before and after denoising.

|  | Accuracy/% | Time Cost/ms | Recall Rate/% | Silhouette Coefficient | F Value | Entropy Value |
|---|---|---|---|---|---|---|
| Before denoising | 77 | 3.5 | 71 | 0.834 | 0.96 | 0.242 |
| After denoising | 96 | 1.92 | 97.14 | 0.928 | 0.96 | 0.237 |

For the classification effect of supervised learning, the signal features before and after denoising are shown in Tables 9 and 10.

**Table 9.** Supervised clustering features of signals before denoising.

| Classification Method | Accuracy/% | Time Cost/ms | Recall Rate | Silhouette Coefficient | F Value | Entropy Value |
|---|---|---|---|---|---|---|
| decision tree | 93.3 | 93.3 | 20 | 0.933 | 0.933 | 0.987 |
| Adaboost | 93.3 | 93.3 | 899 | 0.933 | 0.933 | 0.986 |
| double SVM | 96.0 | 95.8 | 1690 | 0.960 | 0.960 | 0.987 |

**Table 10.** Supervised clustering features of signals after denoising.

| Classification Method | Accuracy/% | Time Cost/ms | Recall Rate | Silhouette Coefficient | F Value | Entropy Value |
|---|---|---|---|---|---|---|
| decision tree | 96.7 | 96.7 | 18 | 0.968 | 0.966 | 0.987 |
| Adaboost | 96.7 | 96.7 | 867 | 0.968 | 0.967 | 0.986 |
| double SVM | 96.1 | 95.8 | 1690 | 0.960 | 0.960 | 0.987 |

From the results obtained in these two tables, it can be seen that for the decision tree algorithm and the AdaBoost algorithm, the accuracy and recall rates are slightly improved. However, for the double SVM algorithm, the accuracy and recall rate did not increase significantly, which shows that for the supervised clustering algorithm, whether denoising has an impact on the clustering accuracy, but the impact is not significant. There is no significant difference in time consumption, indicating that noise does not affect the time complexity of supervised clustering algorithms. The same as the unsupervised clustering algorithm, the F value in the evaluation index indicates that the results of the test sample can represent the true degree of the whole. However, because the supervised algorithm

obtains the classification label in advance, and the denoised samples can better show the characteristics of their own category, the F value increases after denoising.

Combining the tables of unsupervised learning and supervised learning, we can conclude that whether using the clustering method of unsupervised learning or the clustering method of supervised learning, signal denoising can significantly improve classification performance, making the results more accurate and reliable.

## 4. Conclusions

This paper proposes a machine learning fault signal diagnosis and classification model based on wavelet denoising, which can be used for intelligent diagnosis of rolling bearing faults in rotating machinery in practical mechanical applications. Experimental results show that the proposed model is significantly better than traditional diagnostic models on instrument faults. Aiming at weak signal strength and background noise interference, under the quantitative evaluation index, a better noise reduction algorithm is selected for wavelet denoising, which can better restore the true characteristics of the fault signal. Using unsupervised learning and supervised machine learning classification algorithms, the evaluation indicators before and after denoising are compared to make the classification results more accurate and reliable. By designing the fault classification evaluation prediction criteria, the following conclusions are drawn. In equipment fault diagnosis, if the unsupervised machine learning method is used, K-means clustering can achieve good classification results. If the supervised machine learning method is used, the effect of classification by decision tree is the most ideal. This article will help researchers to intelligently diagnose the faults of instruments and equipment.Therefore, the classification model is widely used in industrial production.

In addition, increasing the types of rolling bearing faults, narrowing the characteristic differences of fault signals and improving the adaptability and flexibility of the model will be the focus of our future work. The dataset used in this paper is limited to one hundred groups of two types of faults under different working conditions. More types of faults and larger datasets can be considered in future research to improve the generalization ability of the model. The wavelet denoising algorithm can be further optimized by selecting different wavelet bases, levels of decomposition and threshold functions according to different types of signals. Machine learning models can further improve their performance by tuning parameters, selecting features, or combining multiple models.

**Author Contributions:** Conceptualization, Y.W. and S.F.; methodology, S.F.; software, S.F.; validation, S.F. and R.W.; formal analysis, R.W., Y.W. and S.F.; investigation, Y.W.; resources, R.W.; data curation, R.W.; writing—original draft preparation, S.F.; writing—review and editing, M.M.; visualization, Y.W.; supervision, M.M.; project administration, Y.W. and R.W. All authors have read and agreed to the published version of the manuscript.

**Funding:** This research received no external funding.

**Institutional Review Board Statement:** Not applicable.

**Informed Consent Statement:** Not applicable.

**Data Availability Statement:** Not applicable.

**Conflicts of Interest:** The author declares no conflict of interest.

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
