# Peer review of "A Bearing Fault Diagnosis Method Based on Wavelet Denoising and Machine Learning"

_applsci, doi:10.3390/app13105936_

Round 1

Reviewer 1 Report

Thank to authors for the subject of the study. I would like to say that I have good experience in both electric motors and fault diagnosis using machine learning.

I believe that it would improve the paper scientific soundness if you have added more discussion for research works, not necessarily need to be electric machines, employed machine learning in fault diagnosis. Hence, I would suggest reviewing the following papers and if you have found it relative to this matter feel free to discuss in your paper; (a) https://doi.org/10.1109/TIM.2021.3096869 , (b) https://doi.org/10.1109/TII.2021.3073685 .

It would be good if you could provide a table comparing different methods were used in fault diagnosis using Machine Learning at the end of first section.

I have known there is a review studied different methods in improve electric motors mainly PM motors and provided a very useful table presenting key factors/methods versus each reference. Please review it and try to discuss yours in similar way or even better and feel free to discuss this reference at yours; (c) https://doi.org/10.1109/TMAG.2021.3098392 .  

Presentation comments: 

I recommend authors to present figures more independently via clear caption and legendary as it is not shown in Fig 5 and 6. Readers should be able to get the maximum understanding of a figure just with looking at figure and for further information reading corresponding text in the manuscript. 

Regards, 

One careful review before submitting the final version. 

Reviewer 2 Report

The manuscript is dedicated to the problem of bearing fault diagnosis. As bearings are crucial parts in mechanical devices the problem investigated by the authors is very important from scientific and practical point of view. Authors propose to apply wavelet transform and machine learning methods to diagnose faults of bearings. It is interesting and relatively novel approach.

My comments and suggestions concerning the manuscript are following:

1.      Quality of the diagram in Figure 1 is quite poor, please improve it.

2.      Line 172 – “train weak classifier hi” – probably it should be “high”

3.      Line 243 - four layers of decomposition are chosen. – we usually use the term “levels of decomposition” not “layers”.

4.      Section 2.1 and 2.2 – I think it is noteworthy to add that when selecting the wavelet type some specific quantitive criteria may be applied (see work https://doi.org/10.17559/TV-20140124110406)

5.      Paragraph 3.2.1 – I don’t understand term “vanishing moments” regarding the wavelets. Could authors explain it?

6.      Figure 2 and 3- the value of the difference between the signals is difficult to notice. My advise is to add diagrams where the difference between the signals will be shown.

7.      Line 319  -“2 time SVM” sounds strange for me. Maybe “double SVM” would be better?

Reviewer 3 Report

This paper was presented a bearing fault type classification and identification model based on wavelet denoising and machine learning. Based on the wavelet denoising algorithm, five of the most widely used machine learning models were applied.

Autors were concluted that if unsupervised machine learning methods are used, K-means clustering can achieve good classification results. Other hand, if supervised machine learning methods are used, decision tree classification can achieve the best results.

After checking the English language and style, the paper should be accepted for publication.

Need checking the English language and style.

Reviewer 4 Report

This work had discussed, the bearing fault diagnosis approach based on wavelet de-noising and five machine learning models. 

1. Revise the abstract by adding a summary of the main experimental results. Add one or two sentences in the abstract to show the vital results of the proposed approach.

2. The major contributions of the proposed approach should be included at the end of the introduction section as bullet points. Also, clearly state the novelty of this work.

3. Consider including a block diagram/architectural diagram of the proposed approach. 

4.  The literature review needs to be enhanced by including the suitable papers published papers in 2022 and 2023 related to Bearing Fault Diagnosis.

5.  Some of the symbols and notations mentioned in this manuscript (including the equations) are not defined. Clarify this point.

6. In figure 1, the text should be rearranged/resized to fit within the decision box.

7. The details of the legend mentioned in figures 2, 3, and 4 are not clear. Kindly enlarge the size of the legend to make it readable. Also, the x-axis and y-axis labels are missing in these diagrams. 

8. In figures 5 and 6, the legend details/labels are missing. 

9. Modify the title of tables 5, 6, 7, 8, 9 and 10, there are some typos in these tables, e.g. Serial Number in table 5.

10. Include a discussion section, elucidating the results obtained in section 3. The analysis of the results is too weak, extensive and detailed analysis of the realized results should be provided.

11.  The conclusion section has to show the impact and insights of this research work. 

Overall, writing can be improved. There are grammatical errors in the manuscript and proof-reading is required.

Round 2

Reviewer 4 Report

Authors have addressed most of the comments.

Moderate English language editing is required.